# Network feature-based phenotyping of leaf venation robustly reconstructs the latent space

**Kohei Iwamasa**[1], **Koji Noshita**[1,2]*

1 Department of Biology, Kyushu University, Fukuoka, Fukuoka, Japan, 2 Plant Frontier Research Center, Kyushu University, Fukuoka, Fukuoka, Japan

* noshita@morphometrics.jp

**Data Availability Statement:** All relevant data and code are available on Zenodo at links https://doi.org/10.5281/zenodo.7070266 and https://doi.org/

## Abstract

Despite substantial variation in leaf vein architectures among angiosperms, a typical hierarchical network pattern is shared within clades. Functional demands (e.g., hydraulic conductivity, transpiration efficiency, and tolerance to damage and blockage) constrain the network structure of leaf venation, generating a biased distribution in the morphospace. Although network structures and their diversity are crucial for understanding angiosperm venation, previous studies have relied on simple morphological measurements (e.g., length, diameter, branching angles, and areole area) and their derived statistics to quantify phenotypes. To better understand the morphological diversities and constraints on leaf vein networks, we developed a simple, high-throughput phenotyping workflow for the quantification of vein networks and identified leaf venation-specific morphospace patterns. The proposed method involves four processes: leaf image acquisition using a feasible system, leaf vein segmentation based on a deep neural network model, network extraction as an undirected graph, and network feature calculation. To demonstrate the proposed method, we applied it to images of non-chemically treated leaves of five species for classification based on network features alone, with an accuracy of 90.6%. By dimensionality reduction, a one-dimensional morphospace, along which venation shows variation in loopiness, was identified for both untreated and cleared leaf images. Because the one-dimensional distribution patterns align with the Pareto front that optimizes transport efficiency, construction cost, and robustness to damage, as predicted by the earlier theoretical study, our findings suggested that venation patterns are determined by a functional trade-off. The proposed network feature-based method is a useful morphological descriptor, providing a quantitative representation of the topological aspects of venation and enabling inverse mapping to leaf vein structures. Accordingly, our approach is promising for analyses of the functional and structural properties of veins.

## Author summary

Leaf venation exhibits diverse network structures among taxa and conservation within taxa, reflecting complex evolutionary processes involving functional, developmental, and

10.5281/zenodo.8020856, and on GitHub at links https://github.com/MorphometricsGroup/iwamasa-2022.

**Funding:** This study was supported by Japan Society for the Promotion of Science (JSPS) KAKENHI Grant Numbers 20H01381, 21K14947, 22H04727, (to K.N.), Japan Science and Technology Agency (JST) MIRAI Grant Number JPMJMI20G6 (to K.N.); Moonshot R&D Grant Number JPMJMS2021 (to K.N.), and Bio-oriented technology Research Advancement InstitusioN (BRAIN) Moonshot R&D Grant Number JPJ009237 (to K.N.). The funders had no role in study design, data collection and analysis, decision to publish, or preparation of the manuscript.

**Competing interests:** The authors have declared that no competing interests exist.

structural constraints. We used network features to characterize hierarchical and complex venation patterns for quantitatively understanding their morphological diversities and constraints. We analyzed 479 non-chemically treated leaves of five species and demonstrated that network features contain sufficient information for species classification. Furthermore, we identified biased distribution patterns in the leaf venation morphospace by characterizing leaf samples from both untreated and cleared leaf images. The biased distribution patterns corresponded to a one-dimensional curve, which was predicted by a theoretical study based on the functional trade-offs between optimizing transport efficiency, construction cost, and robustness to damage. These results improve our understanding of morphological constraints and functional trade-offs shaping divergence in leaf venation. Our approach provides a basis for similar analyses in various fields targeting reticulate networks, which are ubiquitous in nature, including biomimetics, generative design, and microfluidics.

## Introduction

Angiosperms show divergence in leaf vein architectures among species but a typical hierarchical network pattern within clades [1,2]. Dicotyledonous leaves are characterized by hierarchical reticulate venation and relative vein widths in successive orders of the hierarchy [3]. The first- to third-order leaf veins are called major veins. Single or multiple first-order veins run from the petiole towards the leaf apex, with relatively small second-order veins branching at intervals and even smaller third-order veins branching from them. The fourth- and higher-order veins, called minor veins, branch to form a reticulate mesh, resulting in a complex network structure [3–5].

Hierarchical leaf venation contributes to hydraulic conductivity, transpiration efficiency, and tolerance to damage and blockage in veins [6–8]. Leaf venation is often evaluated by the vein length per unit area (VLA; also known as the vein density) and is associated with functional traits. VLA is highly correlated with leaf hydraulic conductance ($K_{leaf}$) and the amount of gas exchange per leaf area, that is, leaves with higher VLA tend to have higher photosynthetic and transpiration rates [9–11]. In typical angiosperms, minor veins account for more than 80% of the total vein length [12]. The development of minor veins increases the vein density, and angiosperms achieve higher assimilation rates than those of non-angiosperms [10]. Leaf venation is under morphological constraints related to functional demands (e.g., hydraulic conductivity, transpiration efficiency, and tolerance to damage and blockage) and is thought to show a biased distribution in the morphospace (i.e., in the theoretically possible morphological spectrum, wherein a venation architecture would be represented as a point within the space) [13,14]. The robust and high hydraulic efficiency of leaf venation has been a focus of research in several fields, including biomimetics, generative design, and microfluidics [15–18].

Simple morphological measurements, such as length, diameter, branching angles, and areole areas, and their derived statistics are frequently used to quantify various aspects of leaf veins, especially primary and secondary veins [12,19–25]. Recently, several studies have used topological approaches to evaluate the network structure of leaf veins; this is a promising direction because the vein structure is intrinsically represented as a network. Mieyko et al. [26] proposed a method to quantify loop structures by extending the Horton–Strahler ordering system, which analyzes the topological properties of tree structures, such as river networks and dendritic architectures. Loopy networks have been mapped to binary trees by hierarchically

decomposing loops, preserving the original network structures and connectivity [27,28]. Furthermore, leaf phenotypes are helpful not only for understanding functional traits but also for species identification [1]. These analyses are mainly evaluated based on leaf shape and texture [24,25]. With respect to species identification, recent studies have proposed end-to-end solutions using deep neural network (DNN) models, especially those based on convolutional neural networks (CNN), to extract features from leaf images and classify species [29,30].

Although high-quality images are valuable for quantifying vein patterns, most established methods are time-consuming, invasive, and require special equipment. Chemical cleaning of leaves, a common approach, requires 2–4 weeks to remove tissues with aqueous NaOH and stain the veins with a reagent [31,32]. X-ray imaging systems make it possible to acquire leaf vein images quickly but require dedicated facilities [33]. Obtaining images of raw leaf samples with sufficient quality for extracting higher-order veins from the background can be challenging when using a digital single-lens reflex camera (DSLR) and flat-head scanner due to the low-contrast and intricate leaf textures. A system that enables the acquisition of simple, high-resolution, and high-contrast images is required to quantify leaf veins in many specimens.

In this study, we developed a simple high-throughput phenotyping framework for quantifying vein networks from raw leaf specimens (Fig 1). The proposed method involves four processes: (1) capturing an image from a leaf using a digital camera (and backlighting system for untreated leaves), (2) extracting a vein image from a leaf image using DNN-based semantic segmentation, (3) converting the vein image to a graph, and (4) deriving network features from the graph. This method does not require specialized equipment and enables the evaluation of vein structures in a potentially noninvasive manner. Applying this procedure to leaves of five species, we effectively identified species-specific structures and quantitatively evaluated the vein loops. The proposed method can be used to quantify vein structure properties, such as loopiness and its variation, to reconstruct the latent data space (i.e., a low-dimensional distribution of leaf vein structures within the network feature space), even though higher-order veins are extracted more precisely from cleared leaf images than from raw leaf images.

## Materials and methods

### Leaf materials and image acquisition

A total of 479 leaves of five species (99 *Quercus acutissima*, 166 *Zelkova serrata*, 77 *Prunus × yedoensis*, 49 *Morella rubra*, and 88 *Ficus erecta*) were sampled at the Ito Campus of Kyushu University from August 2021 to November 2021.

A simple digitizing workflow, called the backlight illumination method, was developed to obtain high-contrast leaf vein images from sampled leaf materials. A single leaf was placed on a light table (SV530A*STD-1216U; Sinkohsha, Tokyo, Japan) covered with a non-reflective acrylic plate (PZ-BP600; PFU Limited, Ishikawa, Japan) and images were captured using a digital camera (EOS M6 Mark II; Canon, Tokyo, Japan) mounted on a copy stand (SL700; SFC, Tokyo, Japan) (S1A Fig). To calibrate the area and correct for image distortion, two types of markers were placed in each frame: blue square markers for area calibration and ArUco markers [34] for correcting distortion (S1B Fig). ArUco markers were generated using the image-processing library OpenCV (https://opencv.org/). A simple structure of this frame is available for any size. In this study, frames with two different dimensions were used (140 mm × 190 mm and 95 mm × 140 mm), in which eight ArUco markers and eight 1 cm$^2$ blue square markers were placed according to the leaf size. By using the frame with the ArUco and blue square markers, image distortion was easily corrected, the region of interest was cropped, and sizes were defined. Based on the imaging workflow, a dataset containing sampled leaf materials was

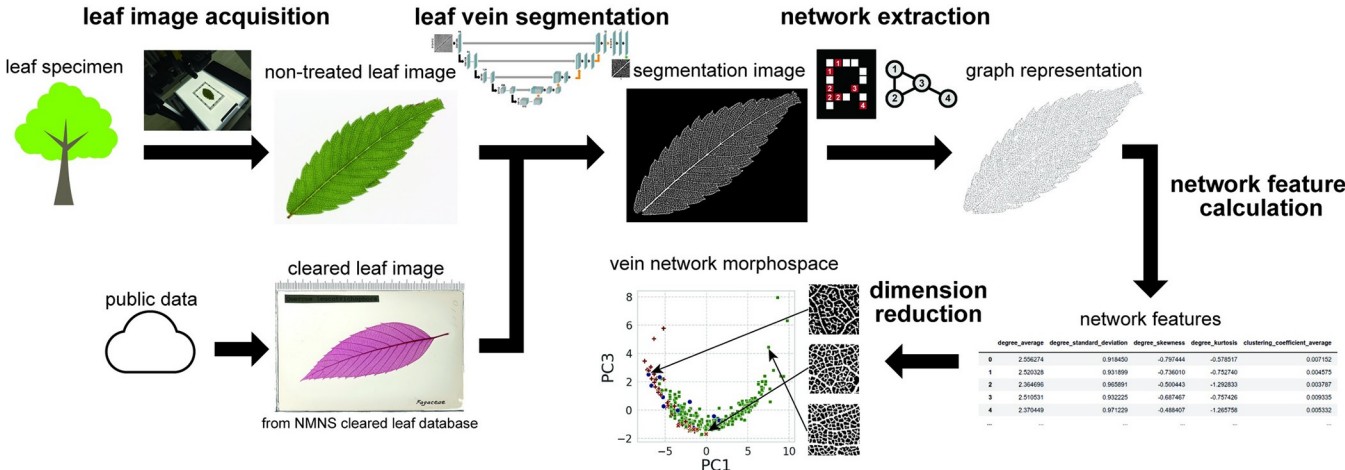

**Fig 1. Overview of the newly developed method for quantifying venation structures.** The method is applicable to both untreated and cleared leaf images. The deep neural network model segmented veins from leaf images. An undirected graph as a representation of the vein network was extracted from the vein image, and network features were used to characterize the venation patterns. By dimensionality reduction, a morphospace was reconstructed based on the observed data.

created (untreated leaf dataset; Fig 2). The images of the untreated leaf dataset have been publicly available at Zenodo [35].

## Cleared leaf image dataset

Two datasets were created based on 4,095 cleared leaf images from the National Museum of Nature and Science (NMNS) Cleared Leaf Database (NMNS, Tokyo, Japan; https://www.kahaku.go.jp/research/db/geology-paleontology/cleared_leaf) [36]. To generate vein images for training the DNN-based semantic segmentation model, 20 high-resolution images were obtained (high-quality dataset; Fig 2).

To compare the results for the untreated leaf dataset with those for cleared leaves, images of the genus corresponding to the sampled species (*Quercus*, *Prunus*, *Zelkova*, *Morella*, and *Ficus*) were filtered from the NMNS Cleared Leaf Database. Some images in the cleared leaf dataset were damaged or showed low resolution; low-quality images were removed by developing a DNN-based image recognition model. First, 200 images were randomly selected and classified into four categories corresponding to quality: *good* (vein structures were easily recognized), *fair* (containing partial noise), *poor* (containing substantial noise), and *very poor* (only partial leaves were captured), based on subjective assessments. The quality of the cleared leaf images was then trained using ResNet18 [37] pre-trained on ImageNet [38]. Using the image recognition model, 328 *good*-quality images were collected (277 *Quercus*, 10 *Zelkova*, 30 *Prunus*, 1 *Morella*, and 10 *Ficus*) (cleared leaf dataset; Fig 2).

## Leaf vein segmentation using U-Net

Leaf veins were extracted from the images using the DNN-based semantic segmentation model, U-Net [39]. The mask images for the training data were created by applying the following image processing procedures to each cleared leaf image in the high-quality dataset without manual annotation (Fig 2A and 2B): (1) Contrast Limited Adaptive Histogram Equalization (CLAHE), which improves the contrast to veins and (2) binarization with a threshold value that was sufficient to binarize the higher-order vein in each image (Fig 2B). Of the 20 images and mask data, 16 were used as a training dataset and 4 as a validation dataset.

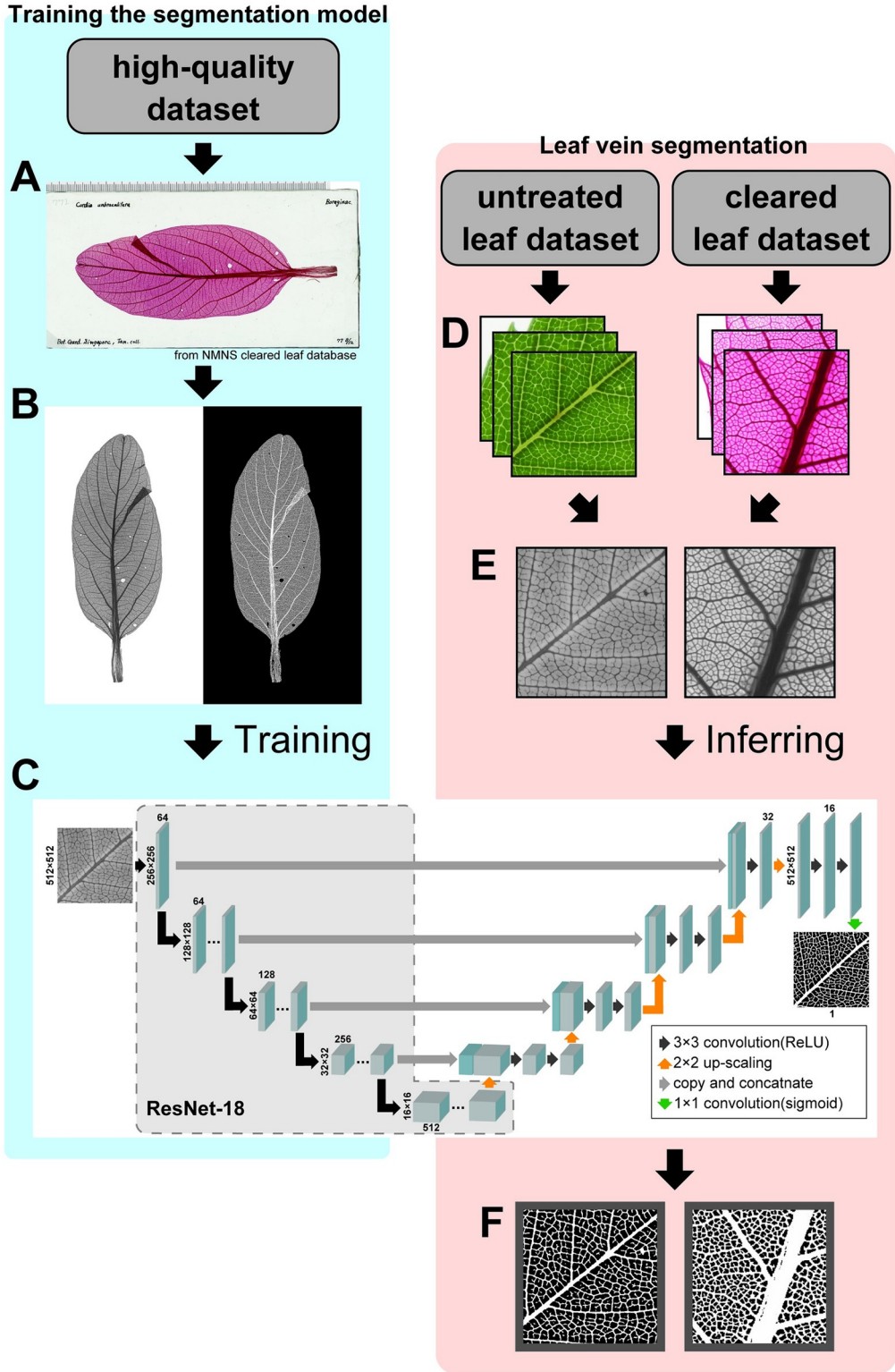

**Fig 2. U-Net for leaf vein segmentation.** Images of the high-quality dataset (A) were converted to grayscale images and masked vein images were generated by conventional image processing with contrast enhancement and binarization (B) to prepare a training dataset. (C) U-Net, a CNN-based model for semantic segmentation, was trained to predict the vein image from the grayscale leaf image. Each blue box corresponds to a feature map. The numbers of channels are shown at the bottom, and the x- and y-sizes are shown on the left. The arrows denote different operations.

(D) To segment the leaf veins, the tiled images of 512 × 512 pixels were obtained from the untreated and cleared leaf dataset. (E) The tiled images were converted to grayscale images. (F) The grayscale images were segmented using the trained U-Net.

To train U-Net on high-resolution images, each image was divided into 512 × 512 pixel tiles, resulting in 768 training images and 177 validation images. The segmentation model used U-Net [39] with ResNet18 [37] pretrained on ImageNet [38] as its encoder block (Fig 2C). A Dice coefficient loss was used for the loss function and Adam was used for the optimizer with a learning rate of 0.001. For training, grayscale images of the leaves were used with data augmentation, that is, rotating the image, changing the gamma value, and adding noise components. Using the trained model, we segmented the veins in images of both untreated and cleared leaf datasets. Each image was divided into 512 × 512 pixels tiles (Fig 2D). Both the cleared and untreated leaf images were converted to grayscale images (Fig 2E). Each grayscale image was segmented into veins and merged into a whole-leaf image. To avoid border effects in each tile, an overlap-tile strategy that excluded the outer 16 pixels during merging was adopted (Fig 2F).

In this study, the segmentation and following steps, including skeletonization, graph representation (**Undirected graph representation of leaf vein networks**), and network feature calculation (**Leaf vein network feature extraction**), were conducted using an Intel(R) Core i9-10900K CPU @ 3.70GHz and an NVIDIA RTX3090 GPU. The analysis code and model weights have been publicly available at Zenodo [40] and GitHub (https://github.com/MorphometricsGroup/iwamasa-2022).

## Undirected graph representation of leaf vein networks

A vein network was represented as an undirected graph in which the nodes and edges correspond to the intersections/termini of veins and their linkages, respectively. To extract undirected graphs from the segmented veins, the following procedure was applied: 1) skeletonization of the vein images, 2) node extraction, and 3) edge detection. First, we obtained a topological skeleton, which is a set of concatenated pixels with a width of one pixel equidistant from the boundary of the leaf vein pixels. The boundary of the leaf vein pixels was defined as the leaf vein pixels of which neighbors include at least one background pixel. Second, the nodes of the vein networks were extracted. If a pixel corresponded to veins and three or more of its 8-neighbor pixels next to each other were veins, the pixel was defined as the branching point; if a pixel corresponded to veins and one of its 8-neighbor pixels was a vein, the pixel was defined as the endpoint (Fig 3A). Adjacent nodes were considered the same node. All independent nodes were indexed to different numbers. The spacing between each branching point was sometimes narrow. To distinguish the independent nodes in the dense vein networks, skeleton images from the cleared leaf dataset were magnified 2× before node extraction. Third, the edges were detected as concatenated pixels between the nodes. We applied a search strategy starting from a randomly chosen node with iterative steps to each 8-neighbor pixel corresponding to the veins. When another node is reached, the iteration is terminated, and the edge connects the two nodes (Fig 3B). For multiple networks, that is, disconnected graphs, extracted from a single image, the connected graph with the highest number of nodes was chosen as the vein network.

## Leaf vein network feature extraction

We used size-invariant network features based on NetSimile [41] to quantify the vein network structure. NetSimile defines network-wide features consisting of several summary statistics of

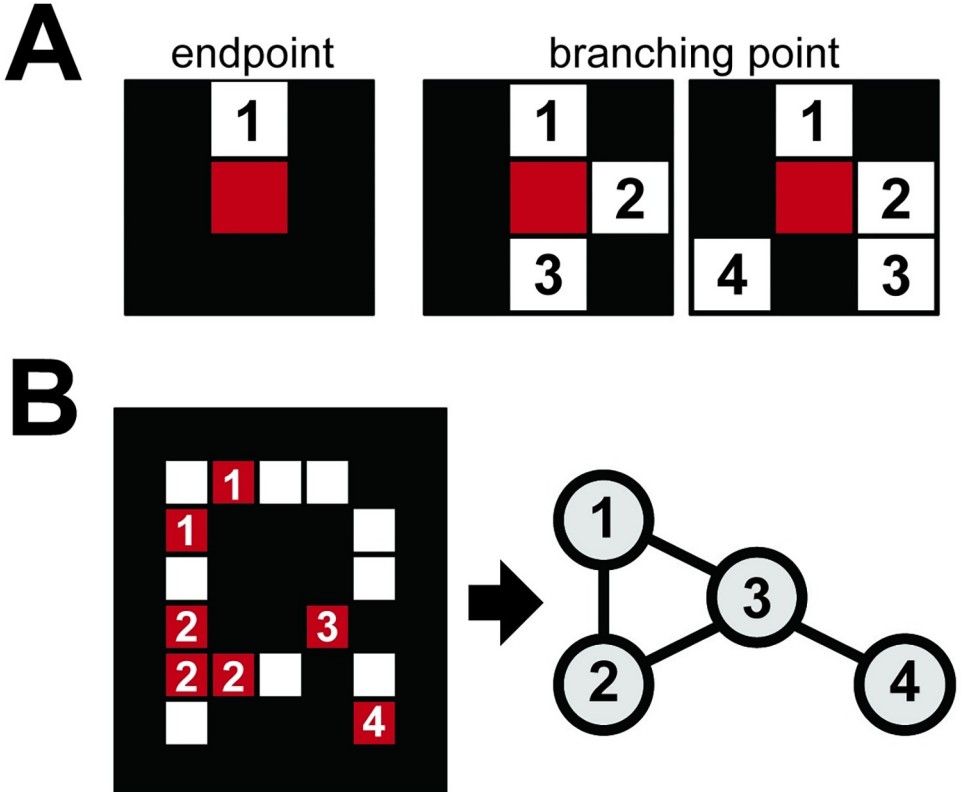

**Fig 3. Undirected graph as a representation of the vein network.** (A) Two types of nodes were detected from the total vein pixels based on the number of neighboring vein pixels; endpoints with one vein pixel in their 8-neighbor pixels and branching points with three or more vein pixels in their 8-neighbor pixels. (B) Adjacent nodes were merged into a single node, and all independent nodes were indexed uniquely. An undirected graph as a representation of the vein network was obtained through an iterative search of edges between nodes.

the features of each node. In this study, the following seven features were calculated for each node: the number of neighbors of a node, clustering coefficient of a node, average number of one-hop neighbors, average clustering coefficient of one-hop away neighbors for a node, number of edges in egonet (one-hop), number of outgoing edges from egonet (one-hop), and number of neighbors of egonet (one-hop). The following four aggregate parameters were also calculated for each feature: average, standard deviation, skewness, and kurtosis. These 28 features were use as the vein network representation. Egonet is "the set of nodes with direct ties to a focal node, called 'ego,' together with the set of ties among members of the ego network" [42]. In this study, egonet is the set of one-hop neighbors of a single node.

## Classification of five species

The random forest model [43] was used to classify five species in the untreated leaf dataset based on 28 network features derived using NetSimile. Accuracy, recall, and precision were evaluated by 5-fold cross-validation on the untreated dataset using the StratifiedKFold class in scikit-learn [44].

## Morphospace analysis of the leaf vein network

We reviewed the morphological diversity of the vein structural networks in an empirical morphospace, that is, the parameter space reconstructed based on the observed data, in which each

point corresponds to the structure of each venation. The morphospace was generated by a principal component analysis (PCA) of the 28 vein network features for the untreated and the cleared leaf datasets. In this study, several morphological properties corresponding to each principal component were identified, and species-specific distribution patterns in the morphospace were observed.

## Results

### Vein structures captured by a simple workflow for digitization

Leaf images of sufficient quality for analyses were successfully captured using the proposed digitizing workflow (Fig 4). The application of extraction steps, that is, segmentation, skeletonization, graph representation, and network feature calculation, to untreated and cleared leaf images took 12.5 and 51.3 seconds per leaf, respectively, using an Intel(R) Core i9-10900K CPU @ 3.70GHz and an NVIDIA RTX3090 GPU. Higher-order veins (i.e., most third- and some fourth-order veins) were recognized in our qualitative observations. The average leaf areas were 33.91 cm$^2$ for *Q. acutissima*, 11.74 cm$^2$ for *Z. serrata*, 30.66 cm$^2$ for *P. × yedoensis*, 13.08 cm$^2$ for *M. rubra*, and 31.56 cm$^2$ for *F. erecta*.

### Undirected graph representation of extracted data from both untreated and cleared leaf images

Although sample preparation differed between cleared and untreated leaf images, the proposed method was able to extract veins with greater clarity than that obtained by simple image processing, while reducing false positives (Fig 4). Skeletonization from the vein images successfully yielded a one-pixel representation of the vein structure while preserving the connectivity and loops. After skeletonization, images were converted to undirected graphs, all of which were indexed and connected (Fig 4).

The numbers of nodes and edges were highly correlated, with correlation coefficients of 0.968 for the untreated leaf dataset and 0.992 for the cleared leaf dataset (Fig 5). Although both the numbers of nodes and edges were also correlated with leaf areas, the numbers of nodes and edges per area of untreated images were underestimated by more than five times compared to those in cleared images (Fig 5B and 5C). Their underestimates could be explained by the inability to clearly distinguish between higher-order veins, especially above the 3rd, from the leaf tissues in the untreated images.

### Species classification based only on network features

The random forest classification using network features for the untreated leaf dataset resulted in an accuracy of 90.6% (Fig 6). The recall rates were 84.7% for *Q. acutissima*, 97.6% for *Z. serrata*, 91.7% for *P. × yedoensis*, 79.6% for *M. rubra*, and 89.4% for *F. erecta*, and the precision estimates were 83.8% for *Q. acutissima*, 97.6% for *Z. serrata*, 85.7% for *P. × yedoensis*, 79.6% for *M. rubra*, and 95.4% for *F. erecta*.

### Empirical morphospace analysis on leaf veins

The empirical morphospace of leaf venation was reconstructed by a PCA based on the network features of the cleared and untreated leaves (Fig 7A). The first three principal components (PC1, PC2, and PC3) of the untreated and cleared leaves explained 88.7% and 91.8% of the total variances, respectively. In the untreated leaf dataset, *Z. serrata* was concentrated in the region with low PC1 values. *F. erecta*, *Q. acutissim*, and *P. × yedoensis* were distributed in regions with high PC1 values but were separated along PC2. In the cleared leaf dataset, *Quercus*

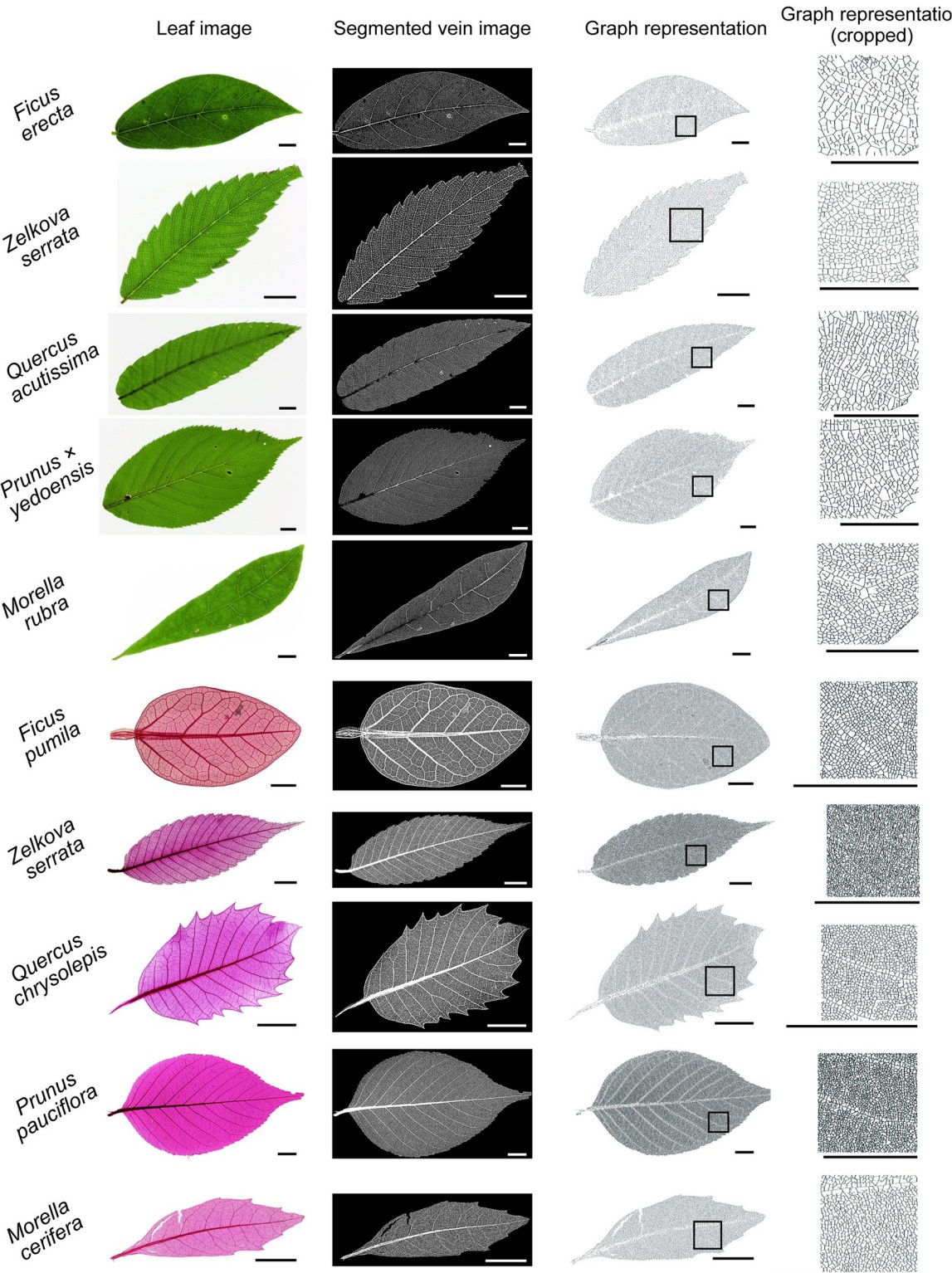

**Fig 4. Examples of vein network extraction.** Leaf vein networks were extracted as undirected graphs from untreated leaf images (*Ficus erecta*, *Zelkova serrata*, *Quercus acutissima*, *Prunus × yedoensis*, and *Morella rubra*; first five rows) and cleared leaf images (*Ficus pumila*, *Zelkova serrata*, *Quercus chrysolepis*, *Prunus pauciflora*, and *Morella cerifera*; last five rows). Cleared leaf images in this figure were based on partially modified images of Specimen Numbers U0604, U4304, T1574, T1426, and U0644 in the NMNS Cleared Leaf Database [36]. Original leaf images, segmented vein images, undirected graphs, and magnified portions correspond to columns. Scale bars represent 10 mm.

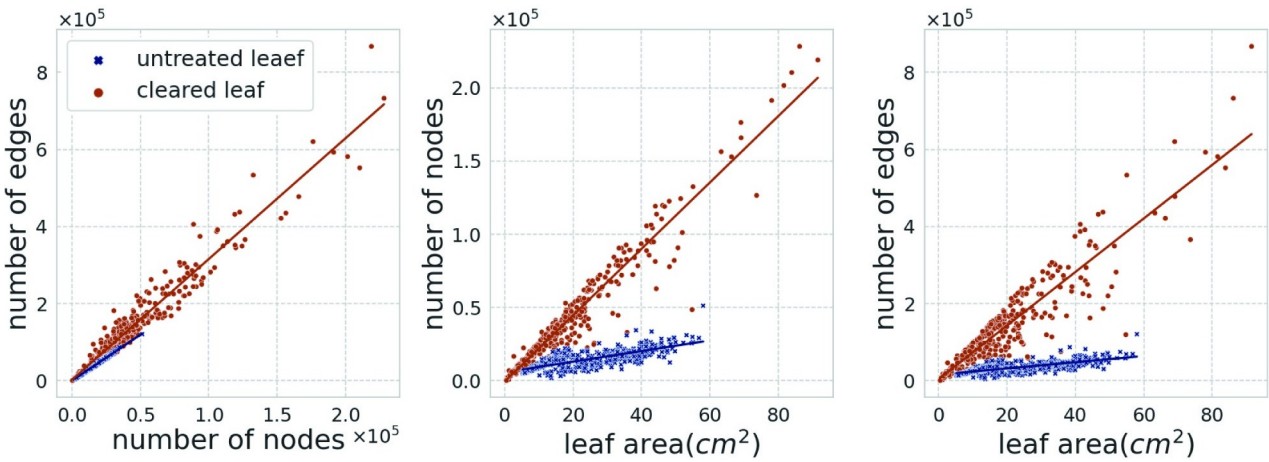

**Fig 5. Scatter plots of nodes, edges, and leaf area.** The slope of each regression line was 2.37 for the untreated and 3.13 for the cleared leaves (number of nodes vs. number of edges); 362.2 for untreated and 2268.5 for cleared leaves (leaf area vs. the number of nodes); 806.3 for the untreated and 6950.1 for the cleared leaf (leaf area vs. the number of edges).

was broadly distributed with PC1 values greater than -5. The other genera were generally distributed in the region with negative scores for PC1, and *Ficus* and *Prunus* showed relatively low and high scores for PC2, respectively.

PC1 represented the loopiness (i.e., the relative frequency of looping structures, known as cycles in graph theory, in a leaf vein network) of higher-order veins (Fig 7B). However, opposite orders for the correspondence of the PC1 axis to the degree of loopiness were observed for

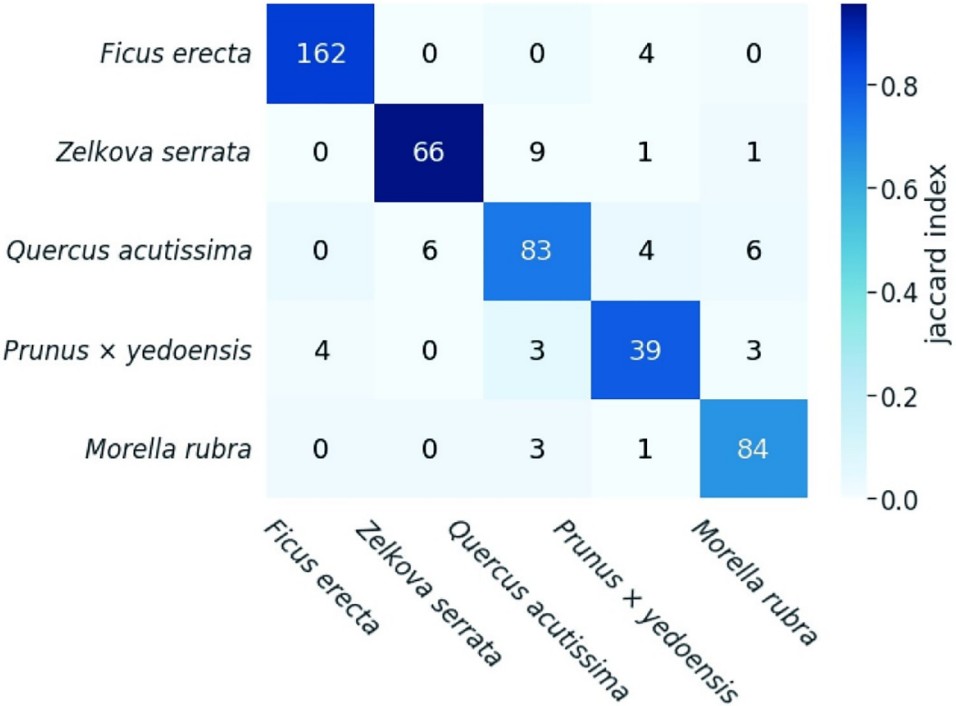

**Fig 6. Confusion matrix for the classification of five species.** Accuracy was 90.6% for the untreated leaf images using the random forest model. The color of each cell corresponds to the Jaccard index.

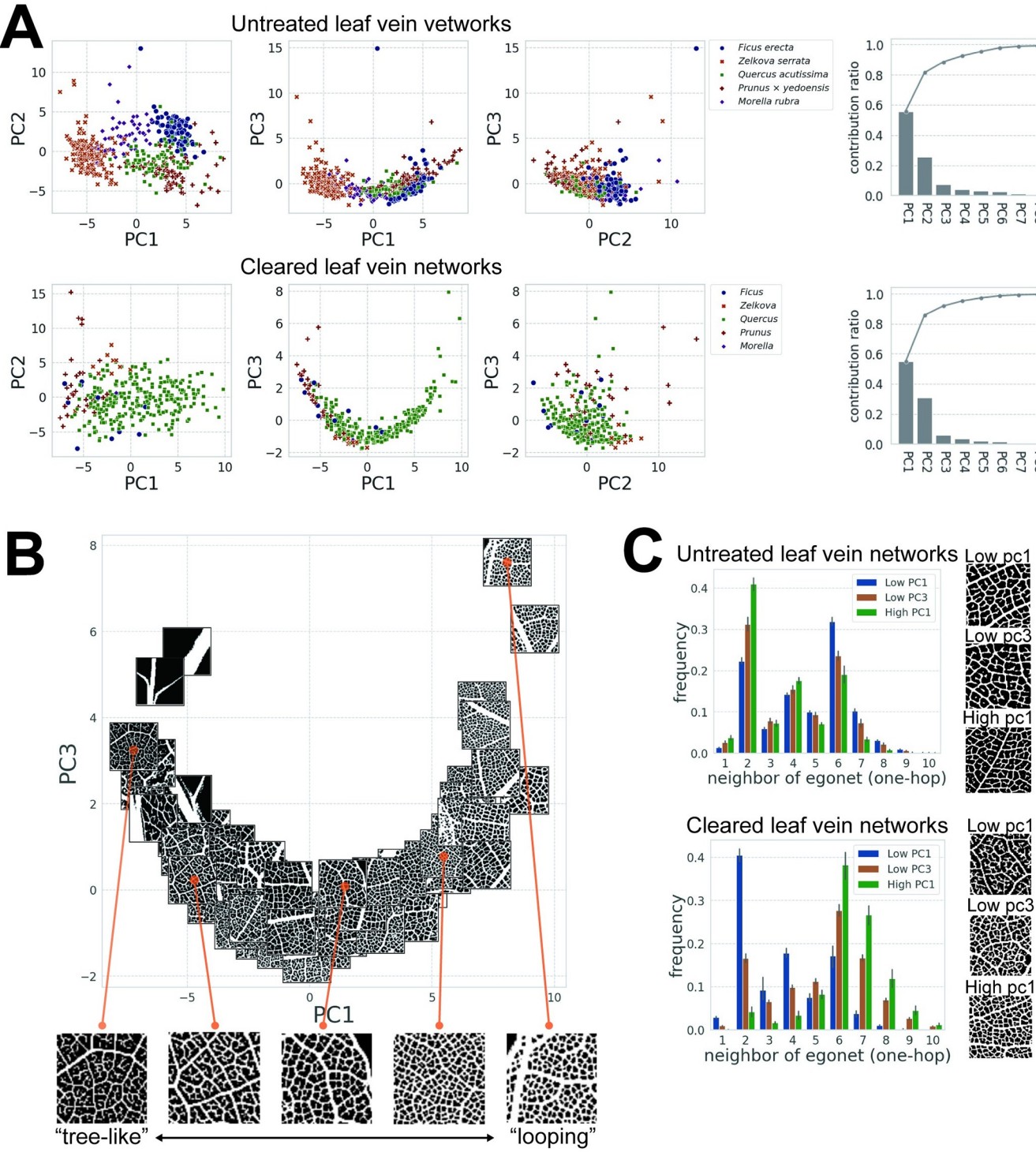

**Fig 7. Empirical morphospace of leaf venation.** (A) Principal component analysis (PCA) of vein network features and contribution rates for the untreated and cleared leaf datasets. Each clade shows a biased distribution, and whole specimens were distributed along one-dimensional U-shaped curves in PC1–PC3 spaces of both datasets. (B) Scatter plot of PC1 and PC3 scores of the cleared leaves with the cropped image of the vein as markers. The degree of loopiness of vein networks changes along a U-shaped curve. (C) Histograms of the mean of the neighborhood of egonet (1-hop) of 10 samples for the lowest PC1, lowest PC3, and highest PC1 in the cleared leaf dataset and untreated dataset. Error bars represent the standard deviation.

untreated and cleared leaf datasets; higher PC1 results in lower loopiness for the untreated leaf dataset; conversely, higher PC1 results in higher loopiness for cleared leaf dataset. Specimens in the untreated leaf dataset had few endpoint nodes and many loops in the higher-order veins when the PC1 scores became large, and specimens in the cleared leaf dataset had more loops when the PC1 scores became low (see the distributions of genera in Fig 7A). The network features related to the egonet showed a high factor loading for PC1, with the number of neighbors of the one-hop egonet significantly contributing to the local looping structures in the vein networks. Vein networks were represented as undirected graphs without crossing edges using algorithms for node extraction and edge detection (Fig 3), resulting in a concentration of one edge per node (endpoint nodes; Fig 3A) or three edges per node (intersection node; Fig 3B). In the case of the number of edges per node being one, a one-hop egonet derived from an endpoint node with one neighbor is a subgraph including two nodes, which are the endpoint node and its neighbor; the number of egonet neighbors was concentrated at two for the low-looping veins with many endpoint nodes (Fig 7C Low PC1 of cleared leaf veins, S4A Fig). For veins with three edges per node, the one-hop egonet is expected to possess two, four, or six neighbors. In our observation, the six neighbor nodes frequently appeared (Fig 7C High PC1 of cleared leaf veins, S4B Fig). Thus, the distribution of neighbors of the egonet, especially the mean and skewness, could effectively characterize vein loopiness (Fig 7C).

The PC2 scores of the cleared samples represent the degree of noise in the images. Some false-positive vein regions (i.e., artifacts recognized as veins in the segmentation step) generated dense networks after skeletonization (S2 Fig). The dense network was characterized by features associated with clustering coefficients.

In the PC1–PC3 space of both the cleared and untreated leaf samples, the network-based feature data showed U-shaped distributions (Fig 7A and 7B). For the untreated specimens, the kurtosis of all nodes positively contributed to PC3; for the cleared specimens, the standard deviation of the number of neighbors of the egonet contributed to PC3. Our results showed that the tree-like and looping structures in the vein networks corresponded to the concentrated distributions (i.e., low standard deviation and high kurtosis) around two and six neighbors of the one-hop egonets, respectively, while intermediate structures exhibited broad distributions across both (i.e., high standard deviation and low kurtosis). For the cleared specimens with more loops (i.e., the number of neighbors of the egonet was concentrated at six) or more endpoint nodes (i.e., concentrated at two), the PC3 value tended to be higher. In contrast, the PC3 value was low for vein networks that contained both intersection and endpoint nodes. Therefore, the one-dimensional U-shaped latent space in PC1–PC3 could be explained by the relationship between the degree of vein loops and the distribution of node features.

## Discussion

### Advantages and limitations of the proposed vein quantification method

The proposed method to extract leaf vein structures is powerful enough to extract vein networks from specimens that were not chemically treated, yet simple and fast. The entire process, from image acquisition to network feature calculation on a non-treated leaf specimen, took only several minutes per leaf, compared with several days for the preparation of a cleared specimen [32]. By devising a leaf image acquisition method, quantification in a noninvasive manner is possible, irrespective of the growth stage. Moreover, the method was affordable, required only a light table, a digital camera, and non-reflective glass (S1 Fig), and did not require specific devices [33]. By generating training datasets from *good*-quality images, the U-Net was trained and segmented into leaf veins of sufficient quality with limited annotation effort (Fig 4).

The network features extracted from leaf images contained sufficient information to identify species-specific structures and represent the loops of higher-order veins. We classified leaves into five species with 90.6% accuracy only based on these features (Fig 6). Previous methods for species identification have focused on statistical analyses of simple parameters (e.g., vein length and diameter) and the application of machine-learning approaches to digital data (e.g., images, polygons, and volumes) [12,19–25,29,30]. For instance, Beghin et al. [25] used the leaf shape signature and texture to identify 18 species, with an accuracy of 81.1% [25]; Kadir et al. [24] developed a model for identifying 32 species (Flavia dataset [45]) based on leaf shape, texture, color, and veins, demonstrating an accuracy of 93.75% [24]. The newly proposed method achieved a comparable accuracy using only the network features, although this was demonstrated in relatively small classes. Topological vein features, which have not been the focus of previous studies, are expected to contribute to the evaluation of traits and species identification [28]. The proposed network feature-based method has good properties as a morphological descriptor; it provides a quantitative representation of the topological aspects of venation as well as several features that could be inversely mapped to leaf vein structures (i.e., PC1 corresponds to the neighbors of the egonet, representing vein loops). Although the proposed method cannot reconstruct vein network structures at arbitrary points in the morphospace, the integration of embedding techniques utilizing graph neural networks and generative models [46] holds promise in expanding the method toward inverse analysis throughout the entire morphospace. Accordingly, the method is useful for analyses of the functional and structural properties of veins.

However, there are two problems with the proposed approach: (1) for leaves with less visible minor veins, sufficient veins were not extracted with backlighting, and (2) the connections between veins were sometimes lost due to shadows of other veins. With respect to the first issue, for some leaves, such as those with thick cuticles, higher-order veins were not detected, despite exposure to transmitted light (e.g., leaves of *Rhus succedanea* showed unclear veins from third-order veins, whereas the cleared specimen clearly showed these veins). Even in leaves where higher-order veins were visible, the numbers of nodes/edges per area were approximately three times lower than those in cleared specimens (Fig 5). With respect to the second problem, "connectivity" is critical for extracting the topological information of leaf veins; however, connective regions shaded by major veins had different contrast against transmitted light from the underside and extraction sometimes failed (S3 Fig). These problems may be resolved by restoring vein images. Recent studies have proposed methods for single-image super-resolution and image denoising using deep learning models [47,48], providing a basis for improving image quality.

## Robust reconstruction of morphospace occupation patterns of leaf vein networks

The network features of leaf veins extracted from both the untreated leaf dataset and the cleared leaf dataset captured a similar morphospace. Although the PCA was conducted independently for each dataset, the data in PC1–PC3 space showed similar U-shape distributions, and the PC1 axis represented the loop of the higher-order veins in both datasets (Fig 7A and 7B). Each clade showed a biased distribution in the morphospace. Although *Quercus* in the cleared leaf dataset distributed a broad range of the morphospace because its sample size is larger than others, *Ficus* and *Prunus* had tree-like structures within the vein networks, and *Zelkova* and *Morella* exhibited relatively high loopiness in both datasets (S5 Fig). This correspondence with the empirical morphospace suggests that the proposed method extracts network features that retain clade-specific morphological properties and is applicable to both sampled

and cleared leaves, independent of the scale. The degree of the loopiness of higher-order venation changes along the U-shaped distributions in both datasets, and this latent one-dimensional occupation pattern implies that there are structural and functional constraints on angiosperm venation patterns. In a previous study, Ronellenfitsch and Katifori [13] theoretically demonstrated that network topologies show a one-dimensional Pareto front, which represents a set of network topologies that cannot improve the functionalities without degrading another, as a result of a trade-off between optimizing transport efficiency, construction cost, and robustness to damage. These functional parameters are considered to correspond to essential functional demands of leaf vein networks, such as hydraulic conductivity, transpiration efficiency, and tolerance to damage and blockage in veins [6–8]. Furthermore, they showed that the network topology changes from a reticulate to a tree-like structure along the one-dimensional Pareto front. Our analysis identified one-dimensional distribution patterns in the vein network morphospace using a data-driven approach and further suggested that leaf venation has been balanced according to several functional constraints along the Pareto front as a consequence of evolutionary processes.

To generalize the results to all angiosperms, however, we need to conduct broader taxon sampling; our analysis only included five genera, and the degree to which other clades show a common network structure is unclear. Public datasets (e.g., [49]), high-throughput phenotyping systems, and simple and affordable approaches, such as the method proposed in this study, are expected to pave the way for expanding data sets and improving our understanding of morphospace occupation patterns and relevant constraints.

## Supporting information

**S1 Fig. Image acquisition system for untreated leaves.** (A) A sampled leaf was placed on a light table and in a frame for correction and was covered with a non-reflective acrylic plate. Leaves were captured using a digital camera. (B) The frame had two markers: arUco and blue square markers. The arUco markers were generated using OpenCV to correct for the image angle and distortion. Blue square markers were used to calculate the leaf area. The frame can be adjusted to any size.
(TIF)

**S2 Fig. Segmented vein images and graph representation of leaves of the cleared leaf dataset along PC2.**
(TIF)

**S3 Fig. Cases of failed vein extraction from untreated leaves.**
(TIF)

**S4 Fig. Schematic diagram of typical local vein network structures.** The blue nodes represent endpoint nodes with a degree of one, while the red nodes represent intersection nodes with a degree of three. Nodes with wavy outlines denote the reference node of an egonet, and orange nodes represent neighboring nodes within a one-hop egonet. (A) illustrated an egonet pattern with a neighborhood size of 2, which was relatively prevalent when the reference node was an endpoint node. (B) portrayed another egonet pattern with a neighborhood size of 6, commonly observed when the reference node was an intersection node.
(TIF)

**S5 Fig. Principal component analysis (PCA) of vein network features and representative leaf vein network structures for each clade.** The representative leaf veins for each clade were adopted as the nearest neighbors of the mean of PC scores within the PC1-PC3 space. (A)

represented untreated leaves, while (B) represented cleared leaves.
(TIF)

## Acknowledgments

We thank Atsushi Yabe for his help with accessing cleared leaf images.

## Author Contributions

**Conceptualization:** Koji Noshita.

**Formal analysis:** Kohei Iwamasa.

**Funding acquisition:** Koji Noshita.

**Investigation:** Kohei Iwamasa, Koji Noshita.

**Methodology:** Kohei Iwamasa, Koji Noshita.

**Supervision:** Koji Noshita.

**Visualization:** Kohei Iwamasa.

**Writing – original draft:** Kohei Iwamasa, Koji Noshita.

**Writing – review & editing:** Kohei Iwamasa, Koji Noshita.

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
