## [Decision Letter · Decision Letter 0]

2 Mar 2023

Dear Dr. Noshita,

Thank you very much for submitting your manuscript "Network feature-based phenotyping of leaf venation robustly reconstructs the latent space" for consideration at PLOS Computational Biology.

As with all papers reviewed by the journal, your manuscript was reviewed by members of the editorial board and by several independent reviewers. In light of the reviews (below this email), we would like to invite the resubmission of a significantly-revised version that takes into account the reviewers' comments.

Please carefully apply the clarifications as requested by the reviewers. Also please make sure that the code is available for review.

We cannot make any decision about publication until we have seen the revised manuscript and your response to the reviewers' comments. Your revised manuscript is also likely to be sent to reviewers for further evaluation.

Sincerely,

Roeland M.H. Merks, Ph.D

Academic Editor

PLOS Computational Biology

Natalia Komarova

Section Editor

PLOS Computational Biology

Reviewer's Responses to Questions

**Comments to the Authors:**

Reviewer #1: In their manuscript, “Network feature-based phenotyping of leaf venation robustly reconstructs the latent space”, Iwamasa and Noshita present an analysis pipeline, from image acquisition to graph-based analysis techniques, to morphometrically analyze leaf venation in five angiosperm species.

I very much enjoyed this manuscript and I believe that these approaches are needed in the leaf venation community. As the authors state, the current state-of-the-art is mostly focused on applying statistics to simple parametric measurements (like length, width, numbers of loops, etc) and the use of CNNs and other modern approaches are desperately needed for segmentation. This manuscript addresses all these issues together and I believe it will have a big impact.

My biggest criticism: zenodo links to the github and the data repositories were redacted in the manuscript and I was not able to evaluate the code. This is going to be the most impactful element of this work, and I hope that the fully reproducible workflow, using the cleared leaf database and U-NET for image segmentation especially, is provided upon publication.

Some overall thoughts:

-It was good to see a truthful analysis of the underestimation of edges and nodes in non-chemically treated leaves. This is to be expected, and the method is powerful because it can make use of the public cleared leaf dataset and apply to backlit photos!

-The graph-based analysis is refreshing to see. The paper focuses on the latent space, and example venation images are shown for datapoints in the PCA space. Do the authors think they could somehow derive meaningful eigen-representations of venation networks?

-Related to the above, I wonder if a discussion of other graph-based analysis approaches, like vector representations through node embeddings or graph distances based on the segmented network could be incorporated. Deriving biological meaning from these powerful mathematical representations of graphs will be interesting to look at as more venation networks are analyzed.

I enjoyed this manuscript a lot, great job!

Reviewer #2: the review is uploaded as an attachment

**Have the authors made all data and (if applicable) computational code underlying the findings in their manuscript fully available?**

Reviewer #1: **No: **I am assuming the github and data repositories will be made publicly available, but the zenodo links were redacted in the manuscript I reviewed

Reviewer #2: **No: **

PLOS authors have the option to publish the peer review history of their article (what does this mean?). If published, this will include your full peer review and any attached files.

Reviewer #1: No

Reviewer #2: No
---

## [Decision Letter · Decision Letter 1]

8 Jun 2023

Dear Dr. Noshita,

Thank you very much for submitting your manuscript "Network feature-based phenotyping of leaf venation robustly reconstructs the latent space" for consideration at PLOS Computational Biology. As with all papers reviewed by the journal, your manuscript was reviewed by members of the editorial board and by several independent reviewers. The reviewers appreciated the attention to an important topic. Based on the reviews, we are likely to accept this manuscript for publication, providing that you modify the manuscript according to the review recommendations.

One of the reviewers asks for some final points to be clarified. Also include the Github link to the source code in a revised version of your manuscript.

Sincerely,

Roeland M.H. Merks, Ph.D

Academic Editor

PLOS Computational Biology

Natalia Komarova

Section Editor

PLOS Computational Biology

Reviewer's Responses to Questions

**Comments to the Authors:**

Reviewer #1: The authors have addressed all of my concerns, and the final published manuscript should contain a publicly available github link to the code.

Reviewer #2: The authors have addressed my comments in a satisfactory manner.

Now, I can clearly understand the contents and claims especially those in Fig. 7B and C.

Here are some thoughts after reading the revised version. These are very minor comments.

・L33 a functional trade-off between what and what might be vague by the first reading.

・L278:"loopiness (i.e., the frequency of looping structures, ...)" spatial scale for the frequency might be vague by the first reading.

**Have the authors made all data and (if applicable) computational code underlying the findings in their manuscript fully available?**

Reviewer #1: Yes

Reviewer #2: Yes

PLOS authors have the option to publish the peer review history of their article (what does this mean?). If published, this will include your full peer review and any attached files.

Reviewer #1: No

Reviewer #2: No

Figure Files:

Data Requirements:

Reproducibility:

References:

---

## [Editor Report · Decision Letter 2]

13 Jun 2023

Dear Dr. Noshita,

We are pleased to inform you that your manuscript 'Network feature-based phenotyping of leaf venation robustly reconstructs the latent space' has been provisionally accepted for publication in PLOS Computational Biology.

Best regards,

Roeland M.H. Merks, Ph.D

Academic Editor

PLOS Computational Biology

Natalia Komarova

Section Editor

PLOS Computational Biology

---

## [Editor Report · Acceptance letter]

24 Jun 2023

PCOMPBIOL-D-22-01372R2 

Network feature-based phenotyping of leaf venation robustly reconstructs the latent space

Dear Dr Noshita,

I am pleased to inform you that your manuscript has been formally accepted for publication in PLOS Computational Biology. Your manuscript is now with our production department and you will be notified of the publication date in due course.

With kind regards,

Zsofia Freund
